# Predictors of unwillingness to receive COVID -19 vaccines among Ethiopian Medical students

Dawit Getachew[1]*, Tewodros Yosef[1], Nahom Solomon[1], Melkamsew Tesfaye[1], Eyob Bekele[2]

1 Department of Public Health, School of Public Health, College of Medical Health Sciences, Mizan Tepi University, Tepi, Ethiopia, 2 Department of Surgery, School of Medicine, College of Medical Health Sciences, Mizan Tepi University, Tepi, Ethiopia

* getdawit2011@gmail.com

## Abstract

### Introduction

Medical students are among the potential risky population for the transmission of COVID 19 infections; their willingness to receive COVID 19 vaccine is not well studied. Thus, this study assessed Predictors of Unwillingness to receive COVID -19 vaccines among Ethiopian Medical students.

### Methods

From the 25th of May, 2020 to the 26th of June, 2021, an institution-based cross-sectional study was done at Mizan-Tepi University Teaching Hospital,On 313 medical students from each department and batch were chosen using a multistage sampling approach. A bivariate and multivariable logistic regression were done to identify the predictors of Unwillingness to receive COVID -19 vaccines. Figures, tables, and graphs were used to present the findings. The adjusted odds ratio and its 95% confidence interval were provided.

### Results

In this study 124 (40.7%) 95% CI (35.1,46.4) medical students were not willing to receive the COVID-19 vaccine. And also increased in the age of the student[AOR 0.43, 95% CI; (.29,.63)], Knowledge status about COVID 19 infection transmission, risk factors and control behavior[AOR 1.45, 95% CI; (1.14, 1.85)], perceived susceptibility to COVID 19 infection [AOR 1.70 (1.15, 2.51)], perceived severity of COVID 19 infection[AOR 1.26 (1.01, 1.57)], perceived benefit of COVID 19 vaccine [AOR .58(.38, .88)], positive attitude towards COVID 19 vaccines [AOR .46(.35, .62)], and confidence in safety and efficacy of the vaccine and public authorities decissin in the best interest of the community [AOR 1.93(1.24, 2.99)] were predictors of non-willingness to receive COVID 19 vaccine.

**Data Availability Statement:** All relevant data are within the paper and its Supporting Information files.

**Funding:** The authors received no specific funding for this work.

**Competing interests:** The authors have declared that no competing interests exist.

**Abbreviations:** AOR, Adjusted Odds Ratio; CI, Confidence Interval; HBM, Health Believe Model: SWE South West Ethiopia; TPB, Theory of planned behavior; WHO, World Health Organization.

## Conclusion

Non-willingness to accept the COVID 19 vaccine was predicted by student age, elements in the Health belief model such as anticipated susceptibility, severity, and benefit, and a positive attitude and trust in the vaccine.

## Introduction

Since the start of the COVID 19 pandemic, there have been 249,523,057 cases reported, of whom 5,048,656 have died and 225,965,382 have recovered; according to this report, 366,424 persons have been infected and 6,509 have died [1]. To fight the pandemic several strategies were developed and implemented; vaccination was the most trusted one. In combination with respiratory and personal hygiene and social distancing, COVID 19 vaccination is a promising strategy to tackle the COVID 19 pandemic.

To date, there are around 8 registered vaccines by WHO. The developments of COVID-19 vaccines were similar to all medicine; first, the vaccines were tested in a laboratory, then on human volunteers [2]. However, high COVID-19 Vaccine Hesitancy is becoming a great concern [3]. Vaccine hesitancy refers to delay in acceptance or refusal of vaccines; It is global problem increasingly being recognized which is common in new or newly introduced vaccines and it differs in a different context [4].

In a study done in South Carolina, About 26.1% of students reported they would take COVID-19 vaccines when available [5]. In the USA Almost half of the college students (47.5%) participants reported hesitancy to receive the COVID-19 vaccine [6]. In a Medical University in Riyadh, Saudi Arabia only 66% accepted the vaccine [7]. Medical students frequently rotate between hospital sites and general practice during their clinical placement years and can potentially be a hidden reservoir for COVID-19 [8]. Among medical students in Kazakhstan, Only 2% were vaccinated and 22.4% showed the potential for COVID-19 vaccine acceptance [9]. In India, Vaccine hesitancy was found among 10.6% of the medical and dental students [10]. In Egypt, 46% of medical students had vaccination hesitancy [11, 12].

In Ethiopia, the willingness to receive COVID 19 vaccine among the general population group was reported to be low; only 31.4% were willing to take the COVID-19 vaccine [13]. In Wolaita Sodo town 46.1% willingness to take a COVID-19 vaccine [14].

The majority of non-medical students showed hesitancy towards obtaining COVID-19 vaccines compared to medical students who were more willing, largely owing to their knowledge and understanding of vaccines [15]. In Uganda COVID-19 vaccine acceptability among medical students was 37.3% and vaccine hesitancy was 30.7% [16]. The reasons for hesitancy were different in a different study [17].

Most University students reported their reasons for receiving the COVID-19 vaccine included preventive purposes, a belief in the safety of the vaccines, and the availability of public awareness information regarding the vaccines [18]. In Saudi Arabia, only 66% percent of r participants had taken the vaccine [19]. The most confirmed barriers to COVID-19 vaccination were deficient data regarding the potential and unknown vaccine's adverse effects and insufficient information regarding the vaccine itself [11].

Varied populations and the same population group with various background information have different COVID 19 vaccination predictions. For example, no demographic characteristics were statistically connected with the COVID-19 vaccine's acceptance [18]. However, in another study, being a male and single medical student was linked to vaccine acceptability [16] the field of education [20], year of education [21]. In addition to this, Perceived risk of getting

COVID 19 in the future, and receiving any vaccine in the past 5 years [16]. Not only this Acceptability of the COVID-19 vaccine was strongly associated getting regularly received the flu vaccine [18].

Despite the fact that the pandemic has affected people from many walks of life, medical students are particularly vulnerable because to their housing situation, workplace conditions, and the difficulties of implementing a preventive strategy while on clinical and community attachments. Furthermore, no previous research on this topic has been conducted in Ethiopia. As a result, the Predictors of Unwillingness to Receive COVID-19 Vaccines among Ethiopian Medical Students were studied in this study.

## Material and methods

### Study area and period

The study was conducted at Mizan Tepi University Teaching Hospital from the 25[th] of May, 2020 to the 26[th] of June, 2021. The University is found in the Southwest of Ethiopia, Mizan Aman City, 590 km away from the capital city of Ethiopia.

### Study design

An Institutional based cross-sectional study was done.

### Population

**Source population and study population.** The source populations were all regular Medical students attending Mizan Tepi University in the 2020/2021 academic year. The study populations were all selected regular Medical students attending Mizan Tepi University in the 2020/2021 academic year.

### Inclusion criteria and exclusion criteria

All medical students attending Mizan Tepi University in the 2020/2021 academic year during the data collection period were included. Medical students who were critically ill and unable to respond during data collection were excluded.

### Sample size determination and sampling technique

The sample size was calculated using single proportion formula in EPI Info software; with the following assumption Proportion of willingness tno accept COVID 19 vaccines as 50%, Margin of error of 5%, and $Z\alpha_{/2}$ 95 CL = 1.96 and total number source population 1100. After adjusting for the small source population and potential non-response rate of 10%, the final sample was 313.

$$n = \frac{(Z_{\frac{\alpha}{2}})^2 P(1-P)}{d^2}$$

The total sample size was proportionally allocated to each batch of the six departments namly Public health, Medicine, Nursing, Pharmacy, Medical lab and Midwifery; to select study participants randomly (Fig 1).

### Data collection methods and materials

The data were collected using a pretested, self-administered questionnaire, which was adapted from previous literature [11, 22–24] and was prepared in the English language. The

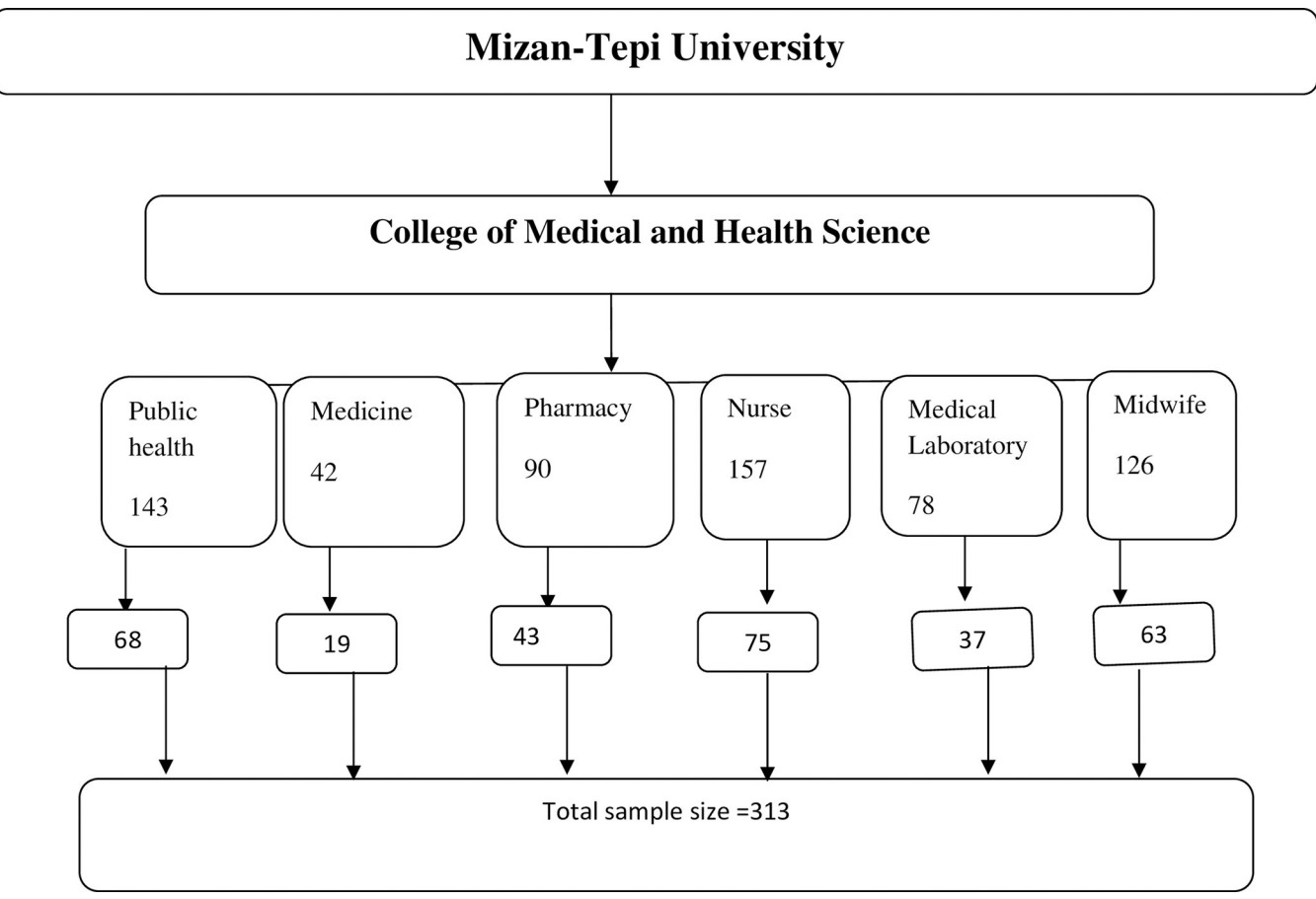

**Fig 1. Sampling distribution flow chart.**

questionnaire has four parts; Part 1 asks about the personal identification information of the student, which includes the age, gender, department, year of the student, and Exposure to COVID-19 infection. Part 2: Knowledge about transmission, risk factors, and preventive behavior of COVID-19 Infection. Part: 3 Items used to measure HBM, The TPB, and The 5C Psychological Antecedents of Vaccination. Part 4; measures the willingness to receive the COVID 19 vaccine [24] and perceived barriers and motivators of COVID-19.

## Data processing and analysis

The data was entered into Epi-Data version 4.2.0 and exported to SPSS version 21 software packages. Descriptive analysis results were presented in the form of tables, figures, frequencies, and summary statistics. The variable which was significant at a p-value less than 0.25 in the bivariate logistic regression analysis was entered into multiple logistic regression, and the variable which has P-value <0.05 was considered a statistically significant predictor.

## Ethics approval and consent to participate

This study was conducted in harmony with the Declaration of Helsinki. Mizan Tepi University Research Ethics Review Committee approved it. All participant were assured the autonomy to participate or not, the a written informed consent was sought from each participant.

## Result

In this study, the response rate is 97.44% as 305 out of 313 samples completed in the interview.

### Personal identification information of the study participants

The mean age of the medical students who participated in this study was 23.22, with a standard deviation of 2.62 and 192. Additionally, 189 of them were men (63%) and 223 (73.1%) of them were from rural areas of the country. In terms of their field, 68 (22.3%) of the students were in public health, 19 (6.2%) in medicine, 43 (14.1%) in pharmacy, 75 (24.6%) were nurses, 37 (12.1%) were in the laboratory, and 60 (20.3%) were midwives. In addition, 22(7.2%) of them have one or more chronic illnesses (Table 1).

### Knowledge about transmission, risk factors, and preventive behavior of COVID-19

The mean score for the overall knowledge about transmission, risk factors, and preventive behavior of COVID-19 was 23.35± 3.86 SD (mean score for Knowledge about COVID-19 transmission was 8.69, ± 2 SD, mean score for Knowledge about COVID-19 risk factors was 6.61 ± 1.80 SD and mean score for Knowledge about COVID-19 Preventive behavior was 8.05 ± 1.07 SD). The overall knowledge level was categorized by taking the mean as the cut of point. Hence, 207(67.9%) of medical student participated in this study have good knowledge about transmission, risk factors, and preventive behavior of COVID-19.

### Willingness to receive COVID-19 vaccine among medical students

In this study, the mean score of willingness to take COVID-19 vaccine among medical student was 8.56 with a 95% CI (8.44, 9.23). When willingness to take COVID-19 vaccine is

**Table 1. Socio-demographic characteristics of study participants.**

| Variable | Category | Frequency n (%) |
|---|---|---|
| Gender | Male | 189(62.0) |
| | Female | 116 (38.0) |
| Religion | Orthodox | 160(52.5) |
| | Protestant | 47(15.4) |
| | Muslim | 98(32.1) |
| Place of origin | Urban | 81(6.9) |
| | Rural | 223(73.1) |
| Department | Public health | 68 (22.3) |
| | Medicine | 19 (6.2) |
| | Pharmacy | 43 (14.1) |
| | Nurse | 75 (24.6) |
| | Medical Laboratory | 37 (12.1) |
| | Midwife | 63 (20.7) |
| Academic Year | 1st year | 34 (11.1) |
| | 2nd Year | 54 (17.7) |
| | 3rd Year | 63 (20.7) |
| | 4th Year | 72 (23.6) |
| | 5th year | 94 (30.) |
| Chronic illness | Yes | 22 (7.2) |
| | No | 283 (92.8) |

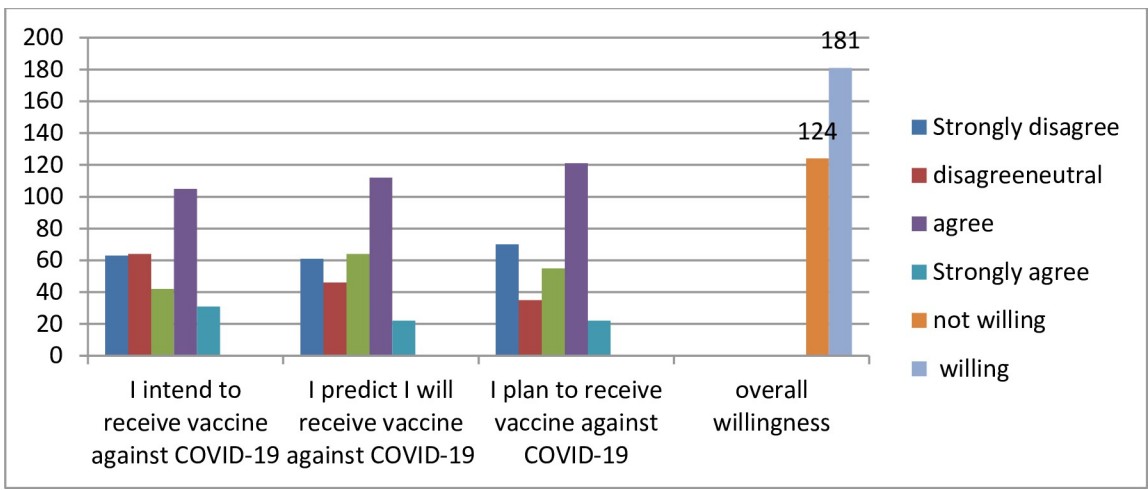

**Fig 2. Willingness to receive vaccine against COVID-19 among medical student.**

dichotomized as below the mean and above the mean, the overall willingness to take the COVID-19 vaccination is 181 (59.3%), while the overall unwillingness to receive the COVID-19 vaccine is 124 (40.7%) with the 95% CI (35.1,46.4).Only 31 (12.1%) of respondents, strongly agreed that they were intended to receive a COVID-19 vaccine, and 22 (8.6%) of respondents highly predicted that they would receive a COVID-19 vaccine in the near future. Likewise, 22 (8.6%) of respondents planned to receive a COVID-19 vaccine (Fig 2).

## Reason for unwillingness to receive COVID 19 vaccine among participants

Medical student who were participated in this study but unwilling to get the COVID 19 vaccine have mentioned the following reasons: vaccine's effectiveness is questioned 60(30.2%), concerned about the vaccine's minimal adverse effects (e.g., fever, pain at the injection site) 59 (29.6), would want to gain more experience 37 (18.6%), I am not afraid of becoming infected with COVID-19.17(8.5%), I am afraid of significant vaccination side effects (e.g., hospitalization, serious illness)15 (7.5%), I am not at risk of serious COVID-19 virus infection 11 (5.5%) (Fig 3).

## Predictors of unwillingness to receive Covid-19 vaccine among medical students

Regarding the factors that affect medical students' willingness to receive COVID 19 vaccine, multivariate analysis showed that only the students' age were significantly associated with their willingness to do so. perceived susceptibility to COVID 19 infection, perceived severity of COVID 19 infection and perceived benefit of COVID 19 vaccine significantly predicted non-willingness to obtain COVID 19 vaccine from the HBM variables, whereas perceived COVID 19 vaccine barriers did not. Additionally, only attitude toward the COVID 19 vaccine was a significant predictor of non-willingness to get the COVID 19 vaccine among the components from the Theory of Planned Behavior. Finally, only confidence has shown a significant association with refusal to take the COVID 19 vaccine among the five C variables.

In this study, for one year increase in the age of medical student, the odds of unwillingness to receive COVID 19 vaccines decreased by 57%, [AOR 0.43, 95% CI; (.29,.63)]. And also, the odds of non-willingness to accept COVID 19 vaccine was 1.45 times higher among medical student who have a high score of knowledge as compared to their counter part [AOR 1.45,

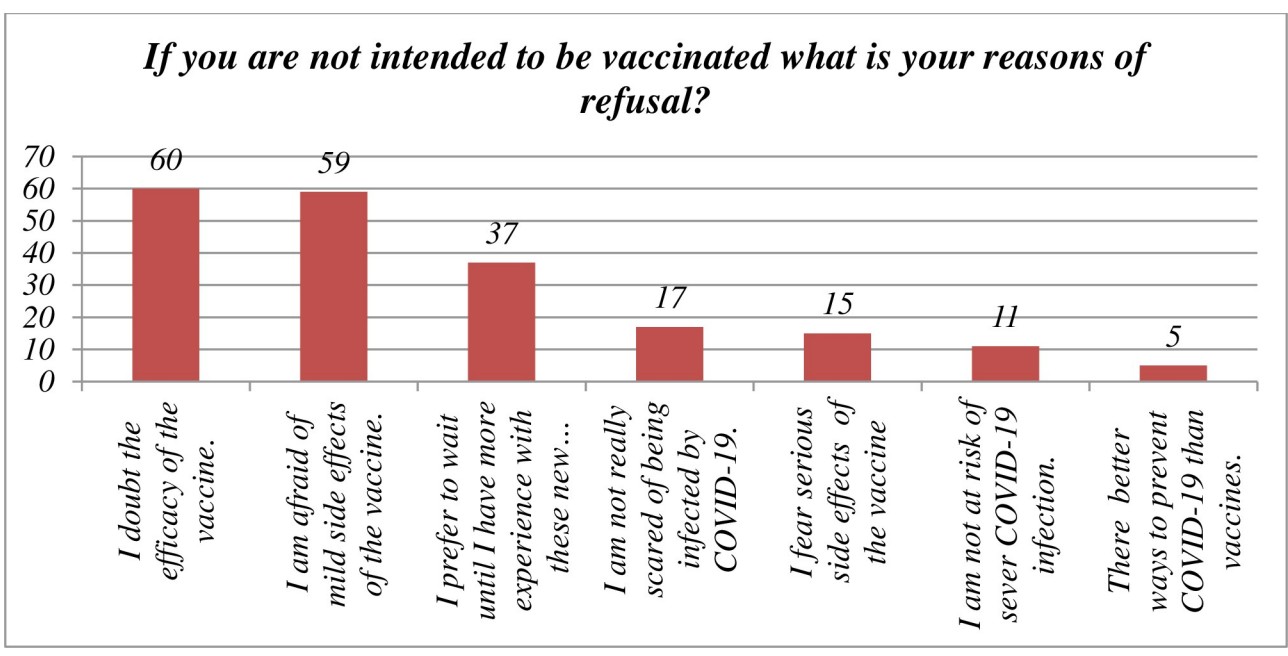

**Fig 3. Reason for none willingness to receive COVID 19 vaccine among medical student.**

95% CI; (1.14, 1.85)]. In addition to this, from the HBM factors perceived susceptibility to COVID 19 infection was 1.70 (1.15, 2.51), perceived severity of COVID 19 infection was 1.26 (1.01, 1.57), and perceived benefit of COVID 19 vaccine was .58(.38, .88) were significantly predicted non-willingness to receive COVID 19 vaccine; while perceived barriers of COVID 19 vaccine were not predicted it. Moreover, from the Theory of planned behavior factors, only attitude towards COVID 19 vaccine was significantly associated with non-willingness to receive COVID 19 vaccine .46(.35, .62). Finally from the 5 C factors, only confidence has shown a statistical association with non-willingness to receive COVID 19 vaccine 1.93(1.24, 2.99). (Table 2).

## Discussion

According to this study, 40.7% with 95% CI (35.1,46.4) medical students were unwilling to recive COVID 19 vaccine., this finding is consistent with a similar study in Uganda [11, 25], but higher than a similar study in Egypt and Kazakhstan [9, 10, 26]. There is also another study that reports higher proportions like china, India [26], and Israel [21, 27–29].

The discrepancy could be explained by the fact that the majority of prior studies were carried out in times of emergency, which might have had an impact on the vast majority of the study participants. As with earlier trials, refusal was motivated by concern over negative consequences as well as doubts about its efficacy [17].

The odds of unwillingness to accept COVID 19 vaccine is 1.45 times higher among medical university students who have higher k knowledge about COVID 19 transmission, risk factors, and preventive behavior.Tthis finding is in line with a similar research report [20, 21]. Because, having the knowledge about the transmission, risk factors, and preventive behavior of COVID 19 infection helps medical student to make appropriate deccession regarding COVID 19 vaccination.

In addition to this, the odds of unwillingness accept COVID 19 vaccine in the studied institution is increased by the three variables in the HBM. Accordingly perceived susceptibility to

**Table 2. Determinants of willingness to receive COVID-19 among medical students in Mizan Tepi University, Ethiopia, 2021.**

| Variable | | Willingness | | COR, 95%CI | COR, 95%CI |
|---|---|---|---|---|---|
| *Personal identification, self-reported health status* | | | | | |
| Gender | Male | 79(63.7) | 110(60.8) | 1 | |
| | Female | 45(36.3) | 71(39.2) | 1.13(.71, 1.82) | |
| Age (μ±SD) | | 23.60±2.79 | 22.95±2.47 | .909(.833, .99)* | .43(.29,.63)** |
| Religion | Orthodox | 67(54.0) | 93(51.4) | 1 | |
| | Protestant | 18(15.5) | 29(16.0) | 1.18(.56. 2.46) | |
| | Muslim | 39(31.5) | 59(32.6) | 1.09(.66, 1.79) | |
| Origin | Urban | 49(39.5) | 33(18.2) | 1 | |
| | Rural | 75(60.5) | 148(81.8) | 2.93(1.74,4.93)* | 1.27(1.04, 1.86)** |
| Filed | Public | 10(8.1) | 9(5.0) | 1 | |
| | Medicine | 23(18.5) | 20(11.0) | .85(.30, 2.35) | |
| | Pharmacy | 22(17.7) | 53(29.3) | .82(.38,1.76) | |
| | Nurse | 14(11.3) | 23(12.7) | 2.27(1.14, 4.52) | |
| | Laboratory | 22(17.7) | 41(22.7) | 1.55(.68, 3.51) | |
| | Midwife | 10(8.1) | 9(5.0) | 1.76(.87, 3.55) | |
| Year of stay (μ±SD) | | 2.94±1.22 | 2.86±1.23 | .95(.79, 1.14) | |
| Chronic | Yes | 7(5.6) | 15(8.3) | 1 | |
| | No | 117(94.4) | 166(91.7) | .66(.56, 1.67) | |
| | Yes | 103(83.1) | 168(92.8) | 1 | 1 |
| | No | 21(16.9) | 13(7.2) | .38(.18, .79)* | .07(.01, .71)** |
| *Knowledge status about COVID 19 infection transmission, risk factors, and control behavior* | | | | | |
| Knowledge (μ±SD) | | 21.40±5.25 | 24.68±1.39 | 1.30(1.19, 1.41)* | 1.45(1.14, 1.85) ** |
| **HBM** | | | | | |
| Susceptibility (μ±SD) | | 4.00±2.22 | 6.70±2.90 | 1.47(1.32, 1.63)* | 1.70(1.15, 2.51)** |
| Severity (μ±SD) | | 4.64±2.89 | 6.83±2.14 | 1.37(1.25, 1.50)* | 1.26(1.01, 1.57) ** |
| Benefit (μ±SD) | | 8.20±2.09 | 9.81±2.13 | 1.43(1.27, 1.60)* | .58(.38, .88)** |
| Barriers (μ±SD) | | 13.87±5.33 | 15.30±5.17 | 1.05(1.00, 1.09)* | 1.10(.97, 1.26) |
| Cues to action (μ±SD) | | 3.26±.93 | 2.87±.81 | .60(.46, .79)* | .63(.35, 1.16) |
| *Theory of planned behavior* | | | | | |
| Attitude (μ±SD) | | 18.75±5.28 | 9.36±3.75 | .56(.49, .62)* | .46(.35, .62)** |
| Subjective norm | | 3.70±1.39 | 3.78±1.09 | 1.06(.87, 1.27) | |
| Behavioral control | | 2.5±0.88 | 2.39±0.81 | 1.04(.79, 1.37) | |
| Regret | | 3.11±1.19 | 3.02±1.08 | .93(.76, 1.14) | |
| *5C psychological antecedents* | | | | | |
| Confidence (μ±SD) | | 8.05±2.51 | 10.06±2.44 | 1.28(1.16, 1.41)* | 1.93(1.24, 2.99)** |
| Constraints (μ±SD) | | 2.89±1.27 | 2.71±1.27 | .89(.75, 1.07) | |
| Complacency (μ±SD) | | 10.05±1.11 | 7.24±2.22 | .89(.75, 1.07) | |
| Calculation (μ±SD) | | 12.20±1.09 | 9.98±.73 | .46(.38, .54) | |
| Responsibility (μ±SD) | | 3.70±1.45 | 8.16±1.53 | - | |

COVID 19 infection increased 1.70, perceived severity of COVID 19 infection increased 1.26, and perceived benefit of COVID 19 vaccine increased .58 times the odds of non-willingness to receive COVID 19 vaccine; while perceived barriers of COVID 19 vaccine were notassociated with unwillingness accept COVID 19 vaccine. This was comparable to prior research, with the exception that in the previous study, all components in the HBM predicted unwillingness to get COVID 19 vaccine, [12, 20, 30]. However in the current study, the perceive barrier did not demonstrate statistical significance association with medical student unwillingness accept

COVID 19 vaccine. This, discrepancey my tell that medical students believe in the studied area with respect to the possible short term and long term side effects of the COVID-19 vaccination did not influenced their status unwilling to accept COVID 19 vaccine. Also, from the variables under Theory of planned behavior only positive attitude toward the COVID 19 vaccine decreased the odds of unwillingness to recive COVID 19 vaccineby 54% [30, 31].

Finaly, from the variables under *5C psychological antecedents only* Confidence is predicted unwillingness of medical student to accept COVID 19 vaccine; that is lack of confidence on safness, effectiveness of COVID 19 vaccines and lack of confidence on public authorities decissin in the best interest of the community increased the odds of unwillingness to accept COVID 19 vaccine 1.93 times among those medical student who lack confidence on COVID 19 vaccine safness and effectiveness. However, three of the 5C psychological antecedents namly Constraints, Complacency Calculation and Responsibility did not significantly predicted medical student unwillingness to accept COVID 19 vaccine. This finding is supported by study finding that studied COVID 19 vaccine hesitancy using HBM, TPB and *5C psychological antecedents* except that in this research constraint did not predicted decision making COVID 19 vaccine [23].

## Strengths and limitation

Although the study's cross-sectional design limits the construction of a causal pathway, it was conducted when the vaccine was accessible, which is an optimal and suggested time to study vaccination reluctance.

## Conclusion

In this study, 40.7% of medical students were not willing to receive the COVID 19 vaccine. And also increasing age, rural origion, perceived susceptibility, the severity of COVID 19 infection and perceived benefit of COVID 19 vaccine, positive attitude about COVID 19 vaccine, and confidence in the safety and efficacy of COVID 19 vaccine were predictors of nonwillingness to receive COVID 19 vaccine. Therefore, the programmer and planner recommended focusing on theoretical constructs to improve vaccine acceptance of medical student.

## Supporting information

**S1 Questionnaire.**
(DOCX)

**S1 Data.**
(SAV)

## Acknowledgments

We would like to acknowledge our data collectors and study participant of this study.

## Author Contributions

**Conceptualization:** Dawit Getachew, Tewodros Yosef.

**Data curation:** Dawit Getachew, Tewodros Yosef, Nahom Solomon.

**Formal analysis:** Dawit Getachew.

**Investigation:** Dawit Getachew.

**Software:** Dawit Getachew.

**Supervision:** Dawit Getachew, Nahom Solomon, Melkamsew Tesfaye, Eyob Bekele.

**Validation:** Dawit Getachew.

**Visualization:** Dawit Getachew, Nahom Solomon, Eyob Bekele.

**Writing – original draft:** Dawit Getachew, Nahom Solomon, Melkamsew Tesfaye, Eyob Bekele.

**Writing – review & editing:** Dawit Getachew, Tewodros Yosef, Nahom Solomon, Melkamsew Tesfaye, Eyob Bekele.

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
