## [Decision Letter · Decision Letter 0]

23 May 2022

PONE-D-21-36100

Predictors of Unwillingness to receive COVID -19 vaccines among Ethiopian Medical student

PLOS ONE

Dear Dr. Getachew,

Thank you for submitting your manuscript to PLOS ONE. After careful consideration, we feel that it has merit but does not fully meet PLOS ONE’s publication criteria as it currently stands. Therefore, we invite you to submit a revised version of the manuscript that addresses the points raised during the review process.

Please submit your revised manuscript in one month. If you will need more time than this to complete your revisions, please reply to this message or contact the journal office at plosone@plos.org. Please include the following items when submitting your revised manuscript:

We look forward to receiving your revised manuscript.

Kind regards,

Subish Palaian

Academic Editor

PLOS ONE

A clean copy of the edited manuscript (uploaded as the new *manuscript* file).

“no fund”

d) If you did not receive any funding for this study, please state: “The authors received no specific funding for this work.

5. Please amend your authorship list in your manuscript file to include author Eyob Bekele.

Additional Editor Comments:

I request authors to revise the manuscript based on reviewers' comments. In addition, the Ethical approval details should be mentioned in the manuscript. The abstract should be formatted as per the journal style. The questionnaire validation should be discussed in detail. The discussion section appears weak and should be rewritten to make stronger arguments and conclusions.

Reviewers' comments:

Reviewer's Responses to Questions

**Comments to the Author**

1. Is the manuscript technically sound, and do the data support the conclusions?

Reviewer #1: Yes

Reviewer #2: Yes

2. Has the statistical analysis been performed appropriately and rigorously? 

Reviewer #1: Yes

Reviewer #2: Yes

3. Have the authors made all data underlying the findings in their manuscript fully available?

Reviewer #1: Yes

Reviewer #2: No

4. Is the manuscript presented in an intelligible fashion and written in standard English?

Reviewer #1: No

Reviewer #2: No

5. Review Comments to the Author

Reviewer #1: The statistical method used is correct. Adequate analyses have been performed. The manuscript is unintelligible in some places. The authors have made all data underlying the findings in their manuscript available if needed.

Reviewer #2: The title is Ok.

ABSTRACTS

There are so many typographical errors . Example: "In this study 145(40%) of medical student were not willing to receive COVID-19 vaccination. And also age of student .43(.29,.63), Knowledge status about COVID 19 infection transmission, risk factors and control behavior 1.45(1.14, 1.85), perceived susceptibility, perceived severity perceived benefit, positive attitude and confidence on safety and efficacy of the vaccine were predictors of non-willingness to receive COVID 19 vaccine."

How can age be .43?

There are so many unnecessary use of upper case. Example: 'Therefore, this study assessed Predictors of Unwillingness to receive COVID -19 vaccines among Ethiopian Medical student. "

INTRODUCTION

The authors should beef up the justification for the study

RESULTS

The authors should include a flow chart showing the participants flow in the study.

6. PLOS authors have the option to publish the peer review history of their article (what does this mean?). If published, this will include your full peer review and any attached files.

Reviewer #1: No

Reviewer #2: No

---

## [Author Response · Author response to Decision Letter 0]

23 Jun 2022

Response: Editors of journals, We've acknowledged the suggestion and updated the manuscript to comply with PLOS ONE's style guidelines. That is the Author, affiliation formatting guide line, which is used to change the title page, and the manuscript body formatting guide line, which is used to change the manuscript body. While doing so, the above-mentioned URL came in handy, and we'd like to thank the journal team for providing these rules.

Response: We have completely copyedited the manuscript for language use, spelling, and grammar, and we appreciate you for your patience.

A clean copy of the edited manuscript (uploaded as the new *manuscript* file

3. Please provide additional details regarding participant consent. 

In the ethics statement in the Methods and online submission information, please ensure that you have specified (1) whether consent was informed and (2) what type you obtained (for instance, written or verbal, and if verbal, how it was documented and witnessed). If your study included minors, state whether you obtained consent from parents or guardians. If the need for consent was waived by the ethics committee, please include this information.

Response: Thank you, editors, for your advice with the ethical statement. We utilized written informed consent. As a result, informed written consent was acquired directly from participants, all of whom were above the age of 18. we have included in the manuscript as follow “This study was conducted in harmony with the Declaration of Helsinki. Mizan Tepi University Research Ethics Review Committee approved it. All participant were assured the autonomy to participate or not, the a written informed consent was sought from each participant.”

“no fund”

d) If you did not receive any funding for this study, please state: “The authors received no specific funding for this work.

Response: The authors received no specific funding for this work

5. Please amend your authorship list in your manuscript file to include author Eyob Bekele.

Response: Thank you, dear editors, for recommending that the authorship list be updated by adding author Eyob Bekele to the text file.

Response: thank you dear editors, we have included the ethical statement in the method section, as follow “This study was conducted in harmony with the Declaration of Helsinki. Mizan Tepi University Research Ethics Review Committee approved it. All participant were assured the autonomy to participate or not, the a written informed consent was sought from each participant.”

Response: Thank you, dear editor; we've added captions to the Supporting Information files at the end of the paper, and we've updated any in-text citations to match.

Additional Editor Comments:

I request authors to revise the manuscript based on reviewers' comments. In addition, the Ethical approval details should be mentioned in the manuscript. The abstract should be formatted as per the journal style. The questionnaire validation should be discussed in detail. The discussion section appears weak and should be rewritten to make stronger arguments and conclusions.

Response: Thank you very much, editor. The majority of the comments from reviewers and editors were accepted. We've also made the necessary changes to the document. The ethical approval statement is included in the manuscript, the abstract part is organized according to journal style, and the validity of the questionnaire was also explored in detail. Finally, we attempted to modify the section on discussion.

Reviewers' comments:

Reviewer's Responses to Questions

Comments to the Author

1. Is the manuscript technically sound, and do the data support the conclusions?

Reviewer #1: Yes

Reviewer #2: Yes

2. Has the statistical analysis been performed appropriately and rigorously?

Reviewer #1: Yes

Reviewer #2: Yes

3. Have the authors made all data underlying the findings in their manuscript fully available?

Reviewer #1: Yes

Reviewer #2: No

 Response: Dear Reviewer, thank you for your time and consideration. the data or supporting material supplied as part of the paper

4. Is the manuscript presented in an intelligible fashion and written in standard English?

 Reviewer #1: No

Reviewer #2: No

 Response: Dear Reviewer, we have updated the English language to meet normal English requirements. 

5. Review Comments to the Author

Reviewer #1: The statistical method used is correct. Adequate analyses have been performed. The manuscript is unintelligible in some places. The authors have made all data underlying the findings in their manuscript available if needed.

Response: Regards, Reviewer We've modified the English language to suit Standard English criteria, making it more understandable.

Reviewer #2: The title is Ok.

ABSTRACTS

There are so many typographical errors . Example: "In this study 145(40%) of medical student were not willing to receive COVID-19 vaccination. And also age of student .43(.29,.63), Knowledge status about COVID 19 infection transmission, risk factors and control behavior 1.45(1.14, 1.85), perceived susceptibility, perceived severity perceived benefit, positive attitude and confidence on safety and efficacy of the vaccine were predictors of non-willingness to receive COVID 19 vaccine."

How can age be .43?

There are so many unnecessary use of upper case. Example: 'Therefore, this study assessed Predictors of Unwillingness to receive COVID -19 vaccines among Ethiopian Medical student. "

Response: thank you dear reviewer we would like to apologize for the typographical errors, we have tried to correct it. We have also corrected unnecessary use of upper cases. 

INTRODUCTION

the authors should beef up the justification for the study

Response: we have tried to add more justification for the study

RESULTS

The authors should include a flow chart showing the participants flow in the study.

Response :

6. PLOS authors have the option to publish the peer review history of their article (what does this mean?). If published, this will include your full peer review and any attached files.

Do you want your identity to be public for this peer review? For information about this choice, including consent withdrawal, please see our Privacy Policy.

Reviewer #1: No

Reviewer #2: No

---

## [Decision Letter · Decision Letter 1]

10 Aug 2022

PONE-D-21-36100R1Predictors of Unwillingness to receive COVID -19 vaccines among Ethiopian Medical studentPLOS ONE

Dear Dr. Getachew,

Thank you for submitting your manuscript to PLOS ONE. After careful consideration, we feel that it has merit but does not fully meet PLOS ONE’s publication criteria as it currently stands. Therefore, we invite you to submit a revised version of the manuscript that addresses the points raised during the review process.

We look forward to receiving your revised manuscript.

Kind regards,

Dragan Pamucar

Academic Editor

PLOS ONE

Journal Requirements:

Reviewers' comments:

Reviewer's Responses to Questions

**Comments to the Author**

1. If the authors have adequately addressed your comments raised in a previous round of review and you feel that this manuscript is now acceptable for publication, you may indicate that here to bypass the “Comments to the Author” section, enter your conflict of interest statement in the “Confidential to Editor” section, and submit your "Accept" recommendation.

Reviewer #1: (No Response)

Reviewer #2: (No Response)

2. Is the manuscript technically sound, and do the data support the conclusions?

Reviewer #1: Partly

Reviewer #2: Yes

3. Has the statistical analysis been performed appropriately and rigorously? 

Reviewer #1: Yes

Reviewer #2: Yes

4. Have the authors made all data underlying the findings in their manuscript fully available?

Reviewer #1: (No Response)

Reviewer #2: Yes

5. Is the manuscript presented in an intelligible fashion and written in standard English?

Reviewer #1: No

Reviewer #2: Yes

6. Review Comments to the Author

Reviewer #1: a. The grammatical errors and English language correction is to be done.

b. The discussion still seems incomplete.

Reviewer #2: The authors should remove the ampersand in 'Study area & period"

The authors should endeavor to add flow chart showing the participants flow.

7. PLOS authors have the option to publish the peer review history of their article (what does this mean?). If published, this will include your full peer review and any attached files.

Reviewer #1: No

Reviewer #2: **Yes: **George Eleje

---

## [Author Response · Author response to Decision Letter 1]

30 Sep 2022

A rebuttal letter that responds to each point raised by the academic editor and reviewer(s)

PONE-D-21-36100R1

Predictors of Unwillingness to receive COVID -19 vaccines among Ethiopian Medical student

Editors comment

PLOS ONE

Dear Dr. Getachew,

Thank you for submitting your manuscript to PLOS ONE. After careful consideration, we feel that it has merit but does not fully meet PLOS ONE’s publication criteria as it currently stands. Therefore, we invite you to submit a revised version of the manuscript that addresses the points raised during the review process.

Author response: Hello reviewer I appreciate the advice and criticism you provided. When the first academic editor left, my colleague and I were concerned. However, now that you are here and have provided such insightful scientific feedback, we are pleased to have you and your help enhance this work as a whole. Additionally, we are ready to consider all suggestions or comments.

Editors comment

Author response: Regarding financial disclosure, as the amount is about equal to our yearly gross income, we have requested publication support.

Editors comment

Author response: Thank you, Editor; we followed the instructions to submit figure files; we found the PACE to be extremely fascinating; we appreciate your advice.

Editors comment

Author response: Thank you, Editor; but I don’t think this is applicable for this manuscript.

We look forward to receiving your revised manuscript.

Kind regards,

Dragan Pamucar

Academic Editor

PLOS ONE

Journal Requirements:

Author response: My dear reviewer, While searching through our reference list, we were unable to locate a retracted article. Inappropriately mentioned articles have been amended and removed, too.

Reviewers' comments:

Reviewer's Responses to Questions

Comments to the Author

1. If the authors have adequately addressed your comments raised in a previous round of review and you feel that this manuscript is now acceptable for publication, you may indicate that here to bypass the “Comments to the Author” section, enter your conflict of interest statement in the “Confidential to Editor” section, and submit your "Accept" recommendation.

Reviewer #1: (No Response)

Reviewer #2: (No Response)

2. Is the manuscript technically sound, and do the data support the conclusions?

Reviewer #1: Partly

Reviewer #2: Yes

3. Has the statistical analysis been performed appropriately and rigorously?

Reviewer #1: Yes

Reviewer #2: Yes

4. Have the authors made all data underlying the findings in their manuscript fully available?

Reviewer #1: (No Response)

Reviewer #2: Yes

5. Is the manuscript presented in an intelligible fashion and written in standard English?

Reviewer #1: No

Reviewer #2: Yes

6. Review Comments to the Author

Reviewer #1: a. The grammatical errors and English language correction is to be done.

Author response: Thank you, reviewer. We have removed any grammatical errors, corrected them, and copy edited the English language

b. The discussion still seems incomplete.

Author response: Reviewer, I appreciate your time. The discussion section has been widened and elaborated where necessary.

Reviewer #2: The authors should remove the ampersand in 'Study area & period"

The authors should endeavor to add flow chart showing the participants flow.

Author response: Please accept our apologies for putting the ampersand in the incorrect place. Thank you. Regarding putting the flow chart we added the flow chart. 

7. PLOS authors have the option to publish the peer review history of their article (what does this mean?). If published, this will include your full peer review and any attached files.

Author response:

Do you want your identity to be public for this peer review? For information about this choice, including consent withdrawal, please see our Privacy Policy.

Reviewer #1: No

Reviewer #2: Yes: George Eleje

Author response: Thank you, Editor; we followed the instructions to submit figure files; we found the PACE to be extremely fascinating; we appreciate your advice.

 ReplyForward

Reviewer comment from the action link "View Attachments"

Summary: The study topic is relevant. The design is overall good but needs some clarifications in certain sections. 

Overall impression: Some doubts have to be cleared.

Author Response: Thank you dear reviewer, you have given us constructive comment we have included your comment in the revised manuscript to clarify certain section. 

Major issues:

Reviewer comment: The semantic error and lack of abbreviations, mostly in the result and discussion sections have made the manuscript difficult to comprehend.

Author Response: Thank you dear reviewer, we apologize for not included list of abbreviation, now in the revised version of the manuscript list of abbreviation was included. 

Reviewer comment: Method section, subsection “Sample Size determination and sampling technique”: Please mention the formula.

Author Response: Thank you dear reviewer, we have mention the formula as single population proportion formula and we have included the formula.

Reviewer comment :Is the questionnaire validated and pre-tested?

Author Response: yes the questionnaire is pretested and validated 

Minor issues:

Reviewer comment: Result section, 349 line number: Is it percentage and number in the bracket? 

Author Response: Thank you dear reviewer, comment accepted the number in the bracket it is percentage and corrected accordingly 207(67.9%).

Reviewer comment: Please elaborate the discussion by adding some contrasting findings to that of other studies.

Author Response: Thank you dear reviewer, we have accepted your comment and elaborated the discussion by adding some contrasting finding.

---

## [Decision Letter · Decision Letter 2]

17 Oct 2022

Predictors of unwillingness to receive COVID -19 vaccines among Ethiopian Medical student

PONE-D-21-36100R2

Dear Dr. Getachew,

We’re pleased to inform you that your manuscript has been judged scientifically suitable for publication and will be formally accepted for publication once it meets all outstanding technical requirements.

Kind regards,

Dragan Pamucar

Academic Editor

PLOS ONE

Additional Editor Comments (optional):

Reviewers' comments:

Reviewer's Responses to Questions

**Comments to the Author**

1. If the authors have adequately addressed your comments raised in a previous round of review and you feel that this manuscript is now acceptable for publication, you may indicate that here to bypass the “Comments to the Author” section, enter your conflict of interest statement in the “Confidential to Editor” section, and submit your "Accept" recommendation.

Reviewer #1: All comments have been addressed

Reviewer #2: All comments have been addressed

2. Is the manuscript technically sound, and do the data support the conclusions?

Reviewer #1: Yes

Reviewer #2: Yes

3. Has the statistical analysis been performed appropriately and rigorously? 

Reviewer #1: Yes

Reviewer #2: Yes

4. Have the authors made all data underlying the findings in their manuscript fully available?

Reviewer #1: Yes

Reviewer #2: Yes

5. Is the manuscript presented in an intelligible fashion and written in standard English?

Reviewer #1: No

Reviewer #2: No

6. Review Comments to the Author

Reviewer #1: (No Response)

Reviewer #2: The authors have addressed the comments raised. The study is now acceptable. COVID-19 remains a topic interests in all concerned.

7. PLOS authors have the option to publish the peer review history of their article (what does this mean?). If published, this will include your full peer review and any attached files.

Reviewer #1: No

Reviewer #2: No

---

## [Editor Report · Acceptance letter]

21 Oct 2022

PONE-D-21-36100R2 

Predictors of unwillingness to receive COVID -19 Vaccines among Ethiopian Medical students 

Dear Dr. Getachew:

I'm pleased to inform you that your manuscript has been deemed suitable for publication in PLOS ONE. Congratulations! Your manuscript is now with our production department. 

Kind regards, 

on behalf of

Dr. Dragan Pamucar 

Academic Editor

PLOS ONE